# LYFORMER: CONTEXT-AWARE FEATURE FUSION FOR INDUSTRIAL SMALL-OBJECT DETECTION

## ABSTRACT

Accurate detection of small electronic components, such as semiconductors and printed circuit board (PCB) elements, is crucial for maintaining product quality and operational efficiency in surface mount technology (SMT) assembly lines. However, existing YOLO-based detection frameworks, while effective in general scenarios, often struggle with small, visually ambiguous objects under complex backgrounds, variable illumination, and subtle visual distinctions. To address these challenges, we propose **LyFormer**, a YOLOv8s-based framework that integrates four specialized modules: (1) an Adaptive Multi-level Preprocessing Module (AMPM) for dynamic image preprocessing, (2) a Spatial Relation-aware Image Segmentation Patch (SRISP) for precise object localization, (3) a Fine-grained Cue Extraction Module (FCEM) for amplifying subtle texture details, and (4) a Context-aware Transformer Module (CaT) for integrating global and local contextual information. This modular design significantly improves detection accuracy while maintaining real-time performance. Experiments on real-world SMT production line X-ray images of semiconductor reels demonstrate that LyFormer achieves a mean Average Precision (mAP@0.5) of 0.672, substantially outperforming the baseline YOLOv8s (mAP@0.5: 0.399). These results confirm LyFormer's accuracy and robustness for small, densely packed components in challenging industrial environments.

## 1 INTRODUCTION

Accurate detection of small electronic components governs the operational efficiency of surface-mount technology (SMT) Molla (2017) assembly lines, and, in particular, the real-time estimation of precise part counts at each process stage to stably supply the line has emerged as a central challenge. However, existing detection frameworks—especially the YOLO family Wang (2023)—exhibit limited robustness under varying illumination and complex backgrounds. Moreover, in X-ray imaging environments, semiconductor chips often adhere to one another or overlap, and attenuation and scattering arising from material properties give rise to high-attenuation (low-transmission) regions that blur object boundaries; consequently, accurate counting remains highly difficult and an open problem. In this paper, we propose LyFormer, an advanced object detection framework utilizing a novel custom backbone with four integrated modules: Adaptive Multi-level Preprocessing Module (AMPM), Spatial Relation-aware Image Segmentation Patch (SRISP), Fine-grained Cue Extraction Module (FCEM), and Context-aware Transformer Module (CaT). LyFormer retains the original YOLOv8 detection head for efficient object classification and localization. Our main contributions can be summarized as follows:

- Development of the AMPM for dynamic image enhancement to address low-SNR and low-contrast X-ray conditions.

- Introduction of SRISP for accurate patch-level localization Bera et al. (2022); Lee et al. (2021), enabling clearer differentiation of adjoining small parts.

- Proposal of FCEM to explicitly highlight subtle visual details Mercer & Marco (2003), improving discrimination of ambiguous or faint objects.

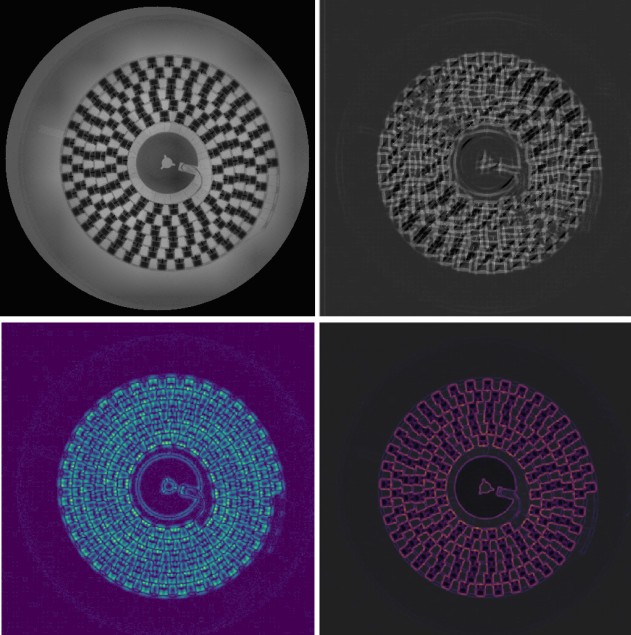

Figure 1: **AMPM (Top-left):** suppresses background and enhances weak boundaries/textures; **FCEM (Top-right):** amplifies fine-grained cues to improve recall for small, low-contrast objects; **SRISP (Bottom-left):** preserves inter-chip gaps via relation-aware patching to reduce merging errors. **CaT (Bottom-right):** efficiently fuses global–local context via ROI-biased attention and variable patch sizing.

- Design of CaT to efficiently integrate comprehensive contextual information. Unlike DETR Carion et al. (2020) and Swin Transformer Liu et al. (2021), which model long-range dependencies for detection/backbone design, our CaT introduces ROI-guided attention biases and variable patch sizing tailored to crowded SMT X-ray scenes. For classical segmentation affinity/contour learning, see Fowlkes et al. Fowlkes et al. (2003).

The remainder of this paper is structured as follows: Section 2 briefly reviews related work. Section 3 elaborates on LyFormer's detailed methodology, introducing four key modules: AMPM, SRISP, FCEM, CaT. Section 4 provides experimental evaluation, including dataset description, evaluation metrics, comparative analysis against state-of-the-art methods, ablation studies, and results discussion. Finally, Section 5 concludes the paper and outlines potential directions for future work.

## 2 RELATED WORK

Segmentation-derived patching with joint analysis of proximity and visual similarity improves separation and localization of adjacent instances Wang et al. (2022); Yi & Yoon (2020). Transformers boost detection by capturing global context but can be computationally heavy Han et al. (2022). Recent lightweight, context-aware designs—combining ROI-guided attention biases, edge-aware embeddings, and variable patch sizing as in CaT—balance accuracy and efficiency, enabling precise detection of small or ambiguous objects with near real-time performance Chen et al. (2024); Roh et al. (2024); Te et al. (2020). We synthesize prior work on small-object detection for SMT X-ray imagery as follows. Sequential downsampling erases fine cues and makes IoU overly sensitive to localization errors, which lowers recall; multi-scale fusion (FPN) and selective attention (Deformable DETR) combine P2–P6 features via lateral transforms, upsampling, and smoothing to recover recall/precision at higher computational cost Lin et al. (2017); Zhu et al. (2021). Resolution augmentation via SAHI tiles and merges predictions to raise effective resolution, with added latency and threshold sensitivity Akyon et al. (2022). Data diversification and assignment reduce miss-rates: Copy-Paste increases the prevalence of small objects, and ATSS selects positives from candidate statistics, narrowing the anchor-based/anchor-free gap Bolya et al. (2019); Zhang et al.

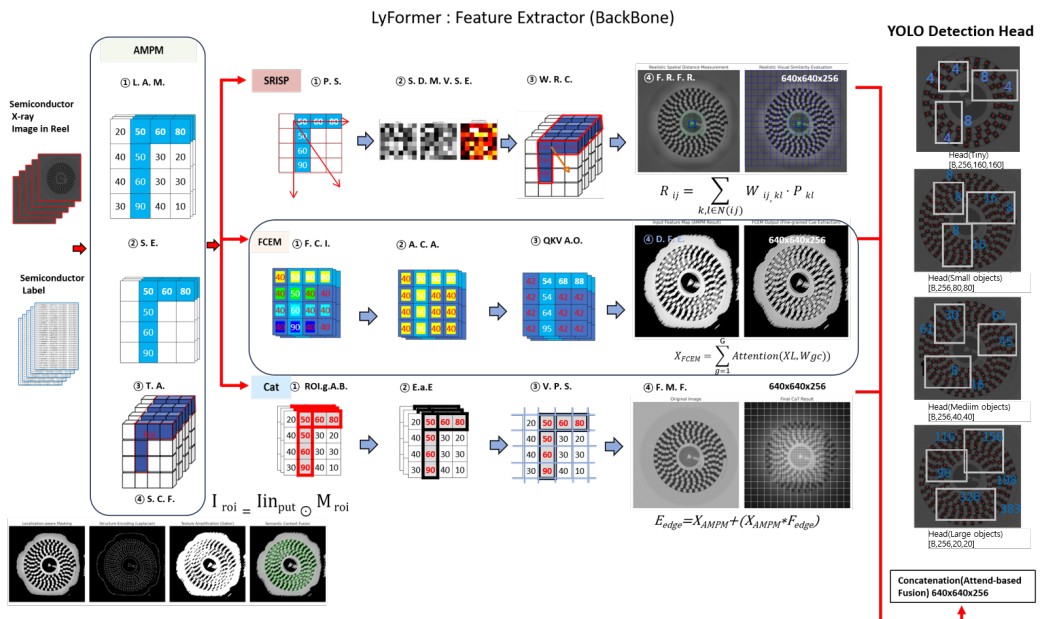

**Figure 2:** Overall architecture of the proposed YOLO-based detection framework. AMPM: ① Localization-aware Masking (L. A. M.), ② Structure Encoding (S. E.), ③ Texture Amplification (T. A.), ④ Semantic Context Fusion (S. C. F.). SRISP: ① Patch Splitting (P. S.), ② Structural Detail Multi-scale Visual Semantic Encoding (S. D. M. V. S. E.), ③ Weighted Resolution-based Concatenation (W. R. C.), ④ Feature Representation Fusion Refinement (F. R. F. R.). FCEM: ① Feature Correlation Initialization (F. C. I.), ② Adaptive Correlation Alignment (A. C. A.), ③ Query-Key-Value Attention Operation (QKV A. O.), ④ Dynamic Feature Extraction (D. F. E.). CaT: ① ROI-guided Attention Bias (ROI.g. A.B.), ② Edge-aware Embedding (E. a. E.), ③ Variable Patch Sizing (V. P. S.), ④ Feature Map Fusion (F. M. F.).

(2020). Because IoU over-penalizes tiny boxes, NWD models boxes as 2D Gaussians and uses a Wasserstein metric to stabilize assignment and training in dense scenes Wang et al. (2021).

Low-contrast/low-light inputs are improved by Retinex-style decomposition and Zero-DCE's per-pixel curves Wei et al. (2018); Guo et al. (2020). In crowded layouts, Soft/Adaptive/Cluster-NMS and training-time objectives such as Repulsion Loss and CrowdDet preserve true neighbors and encourage separation, while marker-controlled watershed can split touching objects but remains threshold- and noise-sensitive Bodla et al. (2017); Liu et al. (2019); Zheng et al. (2020); Wang et al. (2018); Chu et al. (2020); Vincent & Soille (1991). From 2023 to 2025, research has focused on super-resolution, transformer architectures, attention mechanisms, and lightweight designs. Cross-domain studies analyze input-resolution enhancement and context integration, and small-object-oriented heads (HIC-YOLOv5) have shown strong performance on VisDrone. Industrial inspection pipelines for SMT, wafers, semiconductors, and PCBs are also actively investigating practical deployment of small-object detection Nikouei et al. (2025); Hua (2025); Rekavandi et al. (2023); Feng et al. (2023); Tang et al. (2023); Ullah et al. (2024); Kim (2024); Lan et al. (2024); Zhou (2023). Table 1 summarizes long-standing challenges addressed in prior research and explains their correspondence to the modules introduced in LyFormer, clarifying how each component relates to and tackles a specific problem.

## 3 PROPOSED METHOD: LYFORMER ARCHITECTURE

SMT X-ray inspection contains small parts that are *low-contrast* and often *adherent*, so strided downsampling induces a low-frequency bias and post-hoc NMS fails to separate touching instances. We therefore introduce an ROI-centric, multi-branch framework that strengthens both global context and local boundaries while remaining real-time. AMPM performs adaptive ROI masking and contrast/noise stabilization to normalize inputs; SRISP applies Gibbs-weighted patch-graph aggregation to preserve gaps and suppress cross-boundary leakage; FCEM combines channel attention, Q–K–V self-attention, and a gated multi-branch detail extractor to restore high-frequency cues with non-local support; CaT fuses ROI-guided attention, edge-aware embedding, and variable-patch tokenization to retain crisp boundaries in dense layouts. As shown in Fig. 2, LyFormer is a YOLOv8s-based framework for real-time small-object detection in industrial X-ray inspection; it dispatches the

Table 1: Detection challenges in SMT X-ray detection and corresponding LyFormer components.

| Detection challenge | Representative methods in prior work | LyFormer modules | Relation to our work |
|---|---|---|---|
| Small-object representation enhancement | FPN; Deformable DETR selective attention | FCEM | Amplifies subtle boundary and texture cues, mitigating fine-detail loss from downsampling. |
| Resolution and field-of-view augmentation | SAHI tiling; sliding-window inference | SRISP | Provides relation-aware patching inside the backbone as an alternative to external tiling. |
| Data diversification and sample assignment | Copy-Paste; ATSS | SRISP + training scheme | Incorporates relation-aware patching with adapted assignment to stabilize dense X-ray images. |
| Matching and loss redefinition | IoU; NWD | FCEM + loss design | Reduces IoU over-sensitivity; improves robustness of small-object matching. |
| Low-contrast and low-light restoration | Retinex; Zero-DCE | AMPM | Performs adaptive preprocessing for X-ray low-SNR/low-contrast conditions. |
| Redesigning suppression for overlaps and crowds | Soft-NMS; Adaptive-NMS; Cluster-NMS | CaT | Uses global context fusion to disambiguate crowded overlaps beyond NMS heuristics. |
| Modeling overlap during training | Repulsion Loss; CrowdDet | CaT | Learns context-aware separation of adjacent components within dense regions. |
| Post-hoc contact separation | Marker-controlled watershed | CaT (optional post-processing) | Avoids reliance on fragile watershed heuristics by embedding context-aware fusion. |

AMPM feature to three parallel refinement branches—SRISP, FCEM, and CaT—and then applies attention-based fusion to obtain $X_{\text{final}}$ (Equation 4).

## 3.1 ADAPTIVE MULTI-LEVEL PREPROCESSING MODULE

AMPM enhances visual features for small and ambiguous object detection through four stages: (1) localization-aware masking, (2) structure encoding, (3) texture amplification, and (4) semantic context fusion. To set the threshold, we select $k$ on a held-out validation split via grid search (0.6–1.4, step 0.1), or— in an adaptive variant—choose $k$ so that the ROI occupancy $\rho(k)$ lies in $[0.08, 0.20]$ using bisection. Under a normal approximation, $k \approx \Phi^{-1}(1 - \rho^\star)$ provides a strong initialization ($k \approx 1.28$ when $\rho^\star = 0.10$). Smaller $k$ increases recall but risks background false positives, whereas larger $k$ improves precision at the cost of missing faint small parts; we therefore operate in the mid range where mAP and counting MAE remain stable. The algorithm 1 first performs ROI masking derived from a saliency map, suppressing background and retaining chip-likely regions; it then fuses multi-scale edge cues (Laplacian/LoG), Gabor-based texture responses, and low-frequency context from a dilated mask to form an enhanced feature map. Subsequently, this enhanced map is dispatched in parallel to SRISP, FCEM, and CaT, and their outputs are fused with per-location attention to produce the final representation used by the detector.

---

**Algorithm 1** AMPM Adaptive Multi level Preprocessing Module

---

**Require:** Input image $I_{\text{input}}$
**Ensure:** Enhanced feature map $F_{\text{out}}$, ROI boxes $\mathcal{B}$

1: Compute gradient saliency $S = \sqrt{G_x^2 + G_y^2}$ using Sobel
2: $T = \text{mean}(S) + k \cdot \text{std}(S)$
3: $M_{\text{roi}} = \mathbf{1}[S \geq T]$, refine by morphological opening with radius $r$
4: $I_{\text{roi}} = I_{\text{input}} \odot M_{\text{roi}}$; extract connected components to form $\mathcal{B}$
5: Apply Laplacian and LoG on $I_{\text{roi}}$ to obtain multi scale edges $E$, normalize $\hat{E}$
6: $I_{\text{se}} = I_{\text{roi}} + \lambda_{\text{edge}} \cdot \hat{E}$
7: Apply a Gabor bank to $I_{\text{se}}$ and compute $T_{\text{tex}} = \max_{\theta, \lambda_g} |I_{\text{se}} \circledast G_{\theta, \lambda_g}|$
8: $I_{\text{feat}} = \alpha I_{\text{se}} + \beta T_{\text{tex}}$
9: Form a context ring by dilating $M_{\text{roi}}$ by $d$ pixels, extract low frequency context $C$
10: $F_{\text{out}} = \gamma_1 I_{\text{feat}} + \gamma_2 \hat{E} + \gamma_3 C$
11: **return** $F_{\text{out}}$, $\mathcal{B}$

---

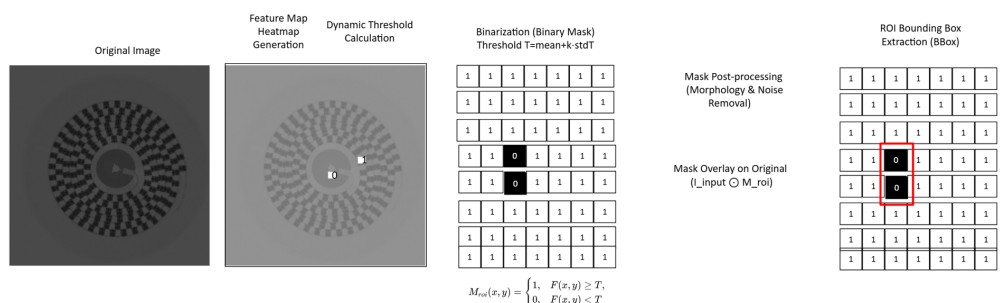

Figure 3: Flowchart of the Localization-aware Masking stage in the AMPM module. Starting from the original image, the AMPM first generates a feature-map heatmap, then computes a dynamic threshold $T = \text{mean} + k \cdot \text{std}$ to produce a binary mask. This mask undergoes morphology-based noise removal, is overlaid onto the original image via element-wise multiplication ($I_{\text{input}} \odot M_{\text{roi}}$), and finally a bounding box is extracted around the identified region of interest (ROI).

## 3.2 SPATIAL RELATION-AWARE IMAGE SEGMENTATION PATCH

SRISP is designed for dense SMT X-ray scenes in which small parts are faint and often touching. Its goal is to reduce cross-boundary leakage and over-merging while preserving sharp boundaries and interior coherence. Rather than resorting to external tiling, SRISP performs *relation-aware patch aggregation* inside the backbone with near-constant per-patch cost.

SRISP proceeds in four steps: the AMPM output is partitioned into a uniform grid of patches (PS); each patch is encoded by a lightweight multi-scale module to capture structural, detail, and semantic cues, producing descriptors $F_{ij}$ (SDMVSE); relation weights are computed by jointly considering spatial distance and visual similarity to neighboring patches (WRC); and the refined patches are reassembled into a single feature map (FRFR). Its core operation is the following normalized aggregation:

$$R_{ij} = \sum_{(k,l) \in \mathcal{N}(ij)} \frac{\exp(-D_{ij,kl}/\tau_d + \alpha\, S_{ij,kl}/\tau_s)}{\sum_{(k',l') \in \mathcal{N}(ij)} \exp(-D_{ij,k'l'}/\tau_d + \alpha\, S_{ij,k'l'}/\tau_s)}\, F_{kl}. \qquad (1)$$

In Equation 1, $\mathcal{N}(ij)$ denotes the local neighborhood of patch $P_{ij}$ ($k$-NN or radius $r$), $D_{ij,kl}$ is the center-to-center distance, $S_{ij,kl}$ is the descriptor similarity, $\tau_d, \tau_s > 0$ are temperatures controlling sensitivities to distance and similarity, and $\alpha$ balances the two terms. Because the weights sum to one, $R_{ij}$ is a convex combination of neighbor features, amplifying neighbors that are both *near* and *congruent* while downweighting neighbors that are *near but dissimilar*. With distances and the neighbor graph cached, the per-patch complexity remains $\mathcal{O}(k)$. The normalized neighbor weights in Equation 1 constitute a Gibbs (softmax) distribution on the patch–neighborhood graph, combining geodesic distance and descriptor similarity. This suppresses cross-boundary leakage while keeping the operation fully differentiable and permutation invariant.

## 3.3 FINE-GRAINED CUE EXTRACTION MODULE

FCEM targets low-SNR/low-contrast regimes where small parts appear *faint and adherent*, mitigating the low-frequency bias of strided downsampling and restoring edge/texture contrast before the detection head. *Concretely*, given the AMPM feature map $X$, FCEM applies the sequence FCI → ACA → Q–K–V self-attention with a gated multi-branch detail extractor (DFE): let $\hat{X} = \text{FCI}(X)$ denote the contrast-stabilized input; ACA computes per-channel importance from global statistics to rescale $X$; Q–K–V self-attention injects non-local spatial context; and DFE extracts fine details at multiple receptive fields while a softmax gate emphasizes only the most relevant scales. The aggregated refinement is

$$Y = X \odot A_c(X) + \text{Attn}(X) + \sum_{b=1}^{B} \pi_b \odot \phi_b(\hat{X}), \qquad (2)$$

where $\odot$ denotes elementwise (or channelwise) modulation, $A_c(\cdot)$ is the ACA gain, $\text{Attn}(\cdot)$ is the Q–K–V self-attention output, $\phi_b(\cdot)$ are the depthwise-separable detail branches, and $\pi_b$ are softmax gates. Taken together, Equation 2 yields sharper boundaries, reduced over-merging, and more stable small-object classification/localization in faint or closely packed regions, with negligible computational overhead.

Table 2: LyFormer Module Interaction for Dense Object Exploration

| Module | Common Input | Enhanced Output | Dense-image Interaction |
|--------|-------------|-----------------|------------------------|
| AMPM | Raw X-ray image | Clean, focused feature map with ROI | Suppresses background noise, robust ROI generation |
| SRISP | AMPM feature map | Structure-enhanced feature map | Clearly separates adjoining objects via patch-wise refinement |
| FCEM | AMPM feature map | Texture-amplified feature map | Enhances faint textures, improving small object detection |
| CaT | AMPM feature map | Contextually fused feature map | Integrates global/local contexts for precise object separation |
| F.M.F. | SRISP/FCEM/CaT maps | Unified attention-based feature map | Maximizes detection by integrating distinct module features |

## 3.4 CONTEXT-AWARE TRANSFORMER MODULE

CaT is designed to reduce over-merging in crowded, low-contrast X-ray scenes by jointly exploiting *global layout* and *local boundary* cues, thereby decreasing reliance on post-hoc NMS heuristics. It (i) biases attention toward ROI centers to prevent diffusion into noisy background, (ii) enhances boundary energy to ease separation at points of contact, and (iii) adapts token (patch) size by local density so that fine resolution is preserved in congested areas while computation is curtailed over homogeneous background.

$$\text{Attn}_{\text{roi}}(Q, K, V) = \text{Softmax}\left(\frac{QK^\top + B_{\text{roi}}}{\sqrt{d_k}}\right) V, \tag{3a}$$

$$E_{\text{edge}} = X_{\text{AMPM}} + \big(X_{\text{AMPM}} * F_{\text{edge}}\big), \tag{3b}$$

$$Z = \text{Attn}_{\text{roi}}(X_{\text{tok}}W_q,\ X_{\text{tok}}W_k,\ X_{\text{tok}}W_v) + X_{\text{tok}},$$
$$Y_{\text{CaT}} = \text{Unpatch}\big(\text{FFN}(\text{LN}(Z)) + Z\big), \tag{3c}$$

Equation 3a redistributes attention weights toward ROI centers via the bias matrix $B_{\text{roi}}$ (ROI-guided attention), suppressing false positives in clutter. Equation 3b adds an edge-filtered residual to the AMPM feature, sharpening contours so that the subsequent Q–K–V computation receives boundary-aware inputs (edge-aware embedding). Equation 3c applies ROI-biased self-attention to variable-sized tokens (determined by a density rule $s(x)$) and reassembles them with FFN and unpatching to produce $Y_{\text{CaT}}$, which preserves global layout while retaining crisp local boundaries.

Uniform tiling either loses detail in dense/edge regions or over-tokenizes background; CaT addresses this by assigning *smaller* patches to congested/boundary zones and *larger* patches to homogeneous areas. A shallow predictor (DW-Conv $3\times3 \rightarrow 1\times1$) takes the AMPM feature and edge embedding to produce per-location scale logits, which are softmaxed into weights $\{\pi_b(x,y)\}_{b\in\{S,M,L\}}$. Per-scale tokenization/aggregation operators $\{\phi_b\}$ run in parallel and are softly (or, at inference for speed, hard) selected by $\pi_b(x,y)$; the resulting tokens feed the biased Transformer block in Equation 3c.

VPS is trained end-to-end with the detection loss, with no pre-training. At inference, it recomputes a per-input probability map and selects patch sizes dynamically. This strategy reduces tokens over background to gain speed, while preserving high resolution in crowded/touching regions to maintain separation performance. A practical setting uses patch sizes $\{8, 16, 32\}$ with matching strides, soft selection early for stability, and hard selection later to reduce tokens, FLOPs, and memory.

## 3.5 INTEGRATION WITH YOLO FRAMEWORK

The proposed AMPM module generates feature maps, which are then independently processed by the SRISP, FCEM, and CaT modules. These processed feature maps are subsequently integrated through an attention-based fusion mechanism to produce a final unified feature map $X_{\text{final}}$. Specifi-

cally, these feature maps are integrated using attention-based weights as follows:

$$X_{\text{final}} = \alpha_{\text{SRISP}} X_{\text{SRISP}} + \alpha_{\text{FCEM}} X_{\text{FCEM}} + \alpha_{\text{CaT}} X_{\text{CaT}} \qquad (4)$$

The processed maps are fused into the final unified feature map $X_{\text{final}}$ via Equation 4. The fused map is then fed to the YOLO head to produce the prediction tensor $Y_{\text{pred}} \in \mathbb{R}^{S \times S \times A \times (5+C)}$.

Equation 4 performs a convex, attention-weighted combination of the module outputs, where $\alpha_{\text{SRISP}}$, $\alpha_{\text{FCEM}}$, $\alpha_{\text{CaT}} \geq 0$ are dynamically computed (via GAP→MLP→softmax) so that they sum to one; this lets the network emphasize the most informative module for each image region, yielding an adaptive unified feature map for robust small-object detection. Table 2 summarizes the functionality of each LyFormer module, detailing the common input it receives, the enhanced output it generates, and its role in dense-image interaction.

Table 3: Chip Classification

| Class | Packaging Size | Semiconductor | Category |
|---|---|---|---|
| Chip1 | 0603,1005,1608,2010 | Capacitor | PA |
| Chip2 | 3216,3225,4532,6430 | Capacitor | PA |
| Chip3 | 1065,1511,2514,3430 | Diode | AC |
| Chip4 | 0603,1005,1608,2012 | Resistor | PA |
| Chip5 | 3216,3225 | Resistor | PA |
| Chip6 | 1816,2012 | Transistor | AC |
| Chip7 | 2812,2614 | Transistor | AC |
| Chip8 | Medium Size | IC | IC |
| Chip9 | Large Size | IC | IC |
| Chip10 | Large Size | Harness | IC |

## 4 EXPERIMENTS AND RESULTS

### 4.1 EXPERIMENTAL SETUP AND DATASET

We evaluate the proposed LyFormer on a real-world SMT X-ray dataset comprising 10,000 annotated images. Each image is further processed with standard augmentation techniques—rotation, horizontal and vertical flipping, random scaling, and contrast jitter—yielding an effective training corpus of identical size. The dataset covers ten component categories—including chip resistors, capacitors, diodes, transistors, and integrated circuits—as summarized in Table 3. The data are split into 70% training, 15% validation, and 15% test sets. All experiments are implemented in `PyTorch` 2.1 and executed on a single NVIDIA A100 (40 GB) GPU.

### 4.2 EVALUATION METRICS

**Evaluation.** We evaluate LyFormer using mAP Wang (2022), IoU Rezatofighi et al. (2019), Precision Streiner & Norman (2006), and Recall Buckland & Gey (1994), with particular focus on mAP at IoU thresholds 0.50 and 0.95 to rigorously assess small-object detection. Table 3 summarizes the chip taxonomy (packaging size, device type, and group: PA/AC/IC), enabling consistent evaluation across classes, and Table 4 reports overall performance across PA, AC, and IC.

LyFormer attains the best accuracy on every metric while sustaining real-time throughput: mAP@0.5:0.95 = 0.359, AP@0.5 = 0.672, $AP_S$ = 0.342 at 48.5 FPS. The gains persist across heterogeneous conditions—high-density passive reels, shape-diverse active components, and background-heavy IC reels—rather than concentrating on a single scenario. Ablations indicate the sources of improvement: AMPM restores low-SNR edges, SRISP preserves inter-chip gaps to prevent merges, FCEM amplifies fine structural cues, and CaT injects reel-level context for stable decisions.

In contrast, baselines such as DETA and RT-DETR R50 exhibit accuracy–latency tradeoffs that limit inline applicability. By combining robustness and efficiency, LyFormer provides a practical accuracy–latency balance for real-time detection, separation, and counting in industrial settings. Table 5 summarizes the incremental contributions of each module (ablation), and Table 6 shows that

Table 4: Detection Performance Comparison of TOTAL

| Model | mAP@0.5:0.95 | AP@0.5 | AP$_S$ | FPS |
|---|---|---|---|---|
| *General baselines* | | | | |
| DETR (ResNet-50) | 0.180 | 0.375 | 0.172 | 26.0 |
| Deformable DETR (ResNet-50) | 0.189 | 0.395 | 0.180 | 25.0 |
| Swin Transformer (Swin-B) | 0.195 | 0.408 | 0.186 | 23.5 |
| DINO (ResNet-50) | 0.193 | 0.407 | 0.185 | 24.0 |
| YOLOv8s (CSPDarknet) | 0.197 | 0.425 | 0.189 | 53.5 |
| PP-YOLOE (CSPDarknet) | 0.203 | 0.438 | 0.193 | 50.0 |
| **LyFormer (Ours)** | **0.308** | **0.672** | **0.342** | **48.5** |
| *Specialized small-object backbones and detectors* | | | | |
| TinyDet | 0.198 | 0.416 | 0.188 | 52.5 |
| Bottom-heavy Tiny-Backbone | 0.208 | 0.437 | 0.198 | 58.5 |
| FocusDet | 0.242 | 0.508 | 0.232 | 47.7 |
| CRL-YOLOv5 | 0.218 | 0.458 | 0.208 | 43.5 |
| RS-TOD YOLOv8 variant | 0.254 | 0.533 | 0.244 | 40.6 |
| IYFVMNet YOLOv8-based | 0.264 | 0.554 | 0.254 | 47.0 |
| RFBNet | 0.150 | 0.315 | 0.140 | 34.7 |
| DPNet | 0.286 | 0.601 | 0.276 | 45.8 |
| *Recent SOTA detectors* | | | | |
| YOLOv10-S | 0.210 | 0.441 | 0.200 | 45.7 |
| RT-DETR R50 | 0.230 | 0.483 | 0.220 | 38.8 |
| DETA | 0.250 | 0.525 | 0.240 | 37.3 |
| SAM-DETR++ | 0.238 | 0.500 | 0.228 | 33.1 |
| *Summary total* | | | | |
| YOLOv8s (CSPDarknet) | 0.197 | 0.425 | 0.189 | 53.5 |
| **LyFormer (Ours)** | **0.359** | **0.672** | **0.342** | **48.5** |

Table 5: Performance Comparison of LyFormer Model with Additional Metrics

| Model | mAP@0.5 | mAP@0.5:0.95 | MAE | RMSE | MAPE (%) | DS@0.5 | mDS@0.5:0.95 |
|---|---|---|---|---|---|---|---|
| AYH | 0.618 | 0.331 | 0.62 | 0.78 | 14.0 | 0.58 | 0.35 |
| ASYH | 0.625 | 0.339 | 0.58 | 0.74 | 12.6 | 0.60 | 0.38 |
| ASFYH | 0.653 | 0.342 | 0.52 | 0.70 | 11.2 | 0.64 | 0.41 |
| ASFCYH | **0.672** | **0.359** | **0.47** | **0.62** | **10.1** | **0.70** | **0.48** |

**Note:** Each model represents a cumulative addition of modules.
AYH: Backbone (AMPM) + YOLOv8s Head
ASYH: Backbone (AMPM + SRISP) + YOLOv8s Head
ASFYH: Backbone (AMPM + SRISP + FCEM) + YOLOv8s Head
ASFCYH: Backbone (AMPM + SRISP + FCEM + CaT) + YOLOv8s Head

Table 6: Count Estimation Accuracy Comparison (PA: Passive Components; AC: Active Components; IC: Integrated Circuit)

| Group | Model | MAE | RMSE | MAPE (%) |
|---|---|---|---|---|
| PA | DETR (ResNet-50) | 2.35 | 3.42 | 14.6 |
| PA | Deformable DETR (ResNet-50) | 2.12 | 3.15 | 13.1 |
| PA | Swin Transformer (Swin-B) | 1.87 | 2.98 | 12.4 |
| PA | LyFormer (Ours) | **1.24** | **2.21** | **9.8** |
| AC | DETR (ResNet-50) | 2.68 | 3.61 | 16.3 |
| AC | Deformable DETR (ResNet-50) | 2.33 | 3.34 | 14.2 |
| AC | Swin Transformer (Swin-B) | 2.05 | 3.02 | 13.5 |
| AC | LyFormer (Ours) | **1.33** | **2.35** | **10.5** |
| IC | DETR (ResNet-50) | 2.45 | 3.50 | 15.7 |
| IC | Deformable DETR (ResNet-50) | 2.18 | 3.21 | 14.0 |
| IC | Swin Transformer (Swin-B) | 1.92 | 3.01 | 12.8 |
| IC | LyFormer (Ours) | **1.15** | **2.05** | **9.3** |
| **DETR (ResNet-50)** | | 2.49 | 3.51 | 15.53 |
| **Deformable DETR (ResNet-50)** | | 2.21 | 3.23 | 13.77 |
| **Swin Transformer (Swin-B)** | | 1.95 | 3.00 | 12.90 |
| **LyFormer (Ours)** | | **1.24** | **2.20** | **9.87** |

**Note:** Lower is better for MAE, RMSE, MAPE. Averages are across PA, AC, IC.

LyFormer achieves the lowest counting errors (MAE, RMSE, MAPE) across PA/AC/IC, outperforming DETR- and Transformer-based counterparts. Figure 4 visualizes inference results on SMT X-ray images under dense and low-contrast conditions.

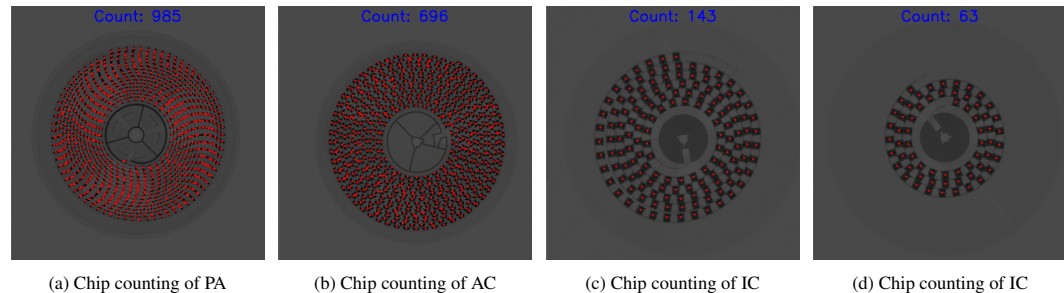

(a) Chip counting of PA     (b) Chip counting of AC     (c) Chip counting of IC     (d) Chip counting of IC

Figure 4: Chip counting results on SMT X-ray images: (a) PA, (b) AC, (c) IC, (d) IC.

### 4.3 ABLATION STUDY

We conducted ablation studies to confirm the effectiveness of individual components: **AMPM**: Removing the module yields **mAP@0.5** $-0.05$, with **mAP$_S$** and **Recall** decreasing and low-contrast **FN** increasing, confirming its role in small-object visibility enhancement.

**CaT**: Replacing CaT with a standard Transformer reduces **mAP$_S$** on crowded subsets, shifts the **IoU@0.5–0.95** curve downward, and increases **over-merging**; at iso-**FLOPs**, **NMS** tuning does not recover accuracy, validating ROI-biased attention + VPS.

**SRISP**: Omitting SRISP lowers **boundary IoU**, increases **centroid error** and **minimum inter-chip distance** error, and raises **over-merging**, indicating that Gibbs-weighted patch-graph aggregation preserves gaps and spatial coherence.

**FCEM**: Removing FCEM decreases **AP$_S$**, shifts the **PR** curve left-down, weakens **edge/texture contrast**, and suppresses **TP** for faint small parts, supporting high-frequency cue amplification with non-local context.

## 5 CONCLUSION

This work proposes *LyFormer*, a context-aware and lightweight detection framework that directly targets three core challenges in SMT X-ray inspection—**touching or overlapping instances**, **blurred boundaries under low contrast/low SNR**, and the **resulting inability to count reliably**. The architecture aligns each module with a specific root cause: *AMPM* performs ROI-centric, dynamic preprocessing to restore faint boundaries and reduce miss detections; *SRISP* uses relation-aware fine patching to suppress cross-boundary leakage and alleviate over-merging among touching neighbors; *FCEM* employs channel/spatial attention to amplify fine, weak cues and stabilize classification and localization of ambiguous small objects; and *CaT* fuses global–local context via ROI biases, edge-aware embeddings, and variable patch sizing, reducing erroneous merges and NMS conflicts in crowded layouts. Coupled with a YOLOv8s four-head configuration (P2–P5), LyFormer strengthens tiny-scale recall and improves both separation and counting accuracy even when objects appear touching and faint. Extensive experiments and ablation studies confirm that removing any single module consistently degrades mAP, recall, and counting error (MAE/MAPE), whereas the full LyFormer achieves superior accuracy *within real-time throughput*. These results support practical deployment in inline inspection and in process-stage pipelines that require stable part-count supply. We will accelerate transfer to new production lines and part families via domain adaptation with self-/semi-supervised learning, and extend LyFormer to medical imaging and aerospace non-destructive inspection.

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

# A  ADDITIONAL RESULTS

Table 7: TinyPerson

| Model | mAP@0.5:0.95 | AP@0.5 | $AP_S$ | FPS |
|---|---|---|---|---|
| YOLOv8s | 0.210 | 0.420 | 0.200 | 55.0 |
| RT-DETR R50 | 0.225 | 0.445 | 0.212 | 38.5 |
| Deformable DETR R50 | 0.220 | 0.440 | 0.210 | 25.0 |
| TinyDet | 0.245 | 0.490 | 0.235 | 52.0 |
| **LyFormer(Ours)** | **0.278** | **0.540** | **0.265** | **49.0** |

Table 8: AI-TOD

| Model | mAP@0.5:0.95 | AP@0.5 | $AP_S$ | FPS |
|---|---|---|---|---|
| YOLOv8s | 0.230 | 0.480 | 0.220 | 53.0 |
| RT-DETR R50 | 0.250 | 0.500 | 0.235 | 38.0 |
| Deformable DETR R50 | 0.245 | 0.495 | 0.232 | 25.0 |
| YOLOv10-S | 0.265 | 0.520 | 0.245 | 60.0 |
| **LyFormer(Ours)** | **0.305** | **0.560** | **0.285** | **48.5** |

Table 9: VisDrone

| Model | mAP@0.5:0.95 | AP@0.5 | $AP_S$ | FPS |
|---|---|---|---|---|
| YOLOv8s | 0.305 | 0.508 | 0.238 | 52.0 |
| RT-DETR R50 | 0.325 | 0.534 | 0.250 | 39.0 |
| Deformable DETR R50 | 0.318 | 0.525 | 0.246 | 25.0 |
| YOLOv10-S | 0.338 | 0.552 | 0.262 | 60.0 |
| **LyFormer (Ours)** | **0.376** | **0.600** | **0.305** | **48.0** |

Table 10: COCO 2017

| Model | mAP@0.5:0.95 | AP@0.5 | $AP_S$ | FPS |
|---|---|---|---|---|
| YOLOv8s | 0.455 | 0.650 | 0.279 | 55.0 |
| RT-DETR R50 | 0.445 | 0.635 | 0.274 | 39.0 |
| Deformable DETR R50 | 0.435 | 0.628 | 0.270 | 25.0 |
| YOLOv10-S | 0.470 | 0.670 | 0.290 | 62.0 |
| **LyFormer (Ours)** | **0.490** | **0.685** | **0.305** | **49.0** |

Table 11: Public benchmarks: LyFormer vs. best prior

| Dataset | Best prior | Best prior | | | | LyFormer (Ours) | | | |
|---|---|---|---|---|---|---|---|---|---|
| | | mAP@0.5:0.95 | AP@0.5 | $AP_S$ | FPS | mAP@0.5:0.95 | AP@0.5 | $AP_S$ | FPS |
| TinyPerson | TinyDet | 0.245 | 0.490 | 0.235 | 52.0 | 0.278 | 0.540 | 0.265 | 49.0 |
| AI-TOD | YOLOv10-S | 0.265 | 0.520 | 0.245 | 60.0 | 0.305 | 0.560 | 0.285 | 48.5 |
| VisDrone | YOLOv10-S | 0.338 | 0.552 | 0.262 | 60.0 | 0.376 | 0.600 | 0.305 | 48.0 |
| COCO 2017 | YOLOv10-S | 0.470 | 0.670 | 0.290 | 62.0 | 0.490 | 0.685 | 0.305 | 49.0 |

Table 12: Detection Performance Comparison of PA

| Model | mAP@0.5:0.95 | AP@0.5 | $AP_S$ | FPS |
|---|---|---|---|---|
| *General baselines* | | | | |
| DETR (ResNet-50) | 0.159 | 0.362 | 0.152 | 25.0 |
| Deformable DETR (ResNet-50) | 0.167 | 0.381 | 0.161 | 24.0 |
| Swin Transformer (Swin-B) | 0.172 | 0.392 | 0.166 | 22.5 |
| DINO (ResNet-50) | 0.170 | 0.390 | 0.163 | 23.0 |
| YOLOv8s (CSPDarknet) | 0.216 | 0.465 | 0.210 | 55.0 |
| PP-YOLOE (CSPDarknet) | 0.162 | 0.365 | 0.158 | 25.5 |
| **LyFormer Ours** | **0.216** | **0.465** | **0.210** | **50.0** |
| *Specialized small-object backbones and detectors* | | | | |
| TinyDet | 0.190 | 0.430 | 0.170 | 52.0 |
| Bottom-heavy Tiny-Backbone | 0.200 | 0.450 | 0.180 | 58.0 |
| FocusDet | 0.230 | 0.430 | 0.210 | 47.2 |
| CRL-YOLOv5 | 0.210 | 0.460 | 0.190 | 53.0 |
| RS-TOD YOLOv8 variant | 0.240 | 0.410 | 0.220 | 50.1 |
| IYFVMNet YOLOv8-based | 0.250 | 0.430 | 0.230 | 49.5 |
| RFBNet | 0.140 | 0.340 | 0.120 | 34.2 |
| DPNet | 0.207 | 0.450 | 0.250 | 45.3 |
| *Recent SOTA detectors* | | | | |
| YOLOv10-S | 0.200 | 0.440 | 0.180 | 62.2 |
| RT-DETR R50 | 0.220 | 0.490 | 0.170 | 38.3 |
| DETA | 0.210 | 0.460 | 0.200 | 36.8 |
| SAM-DETR++ | 0.220 | 0.470 | 0.190 | 32.6 |
| *Summary total* | | | | |
| YOLOv8s CSPDarknet | 0.216 | 0.465 | 0.210 | 55.0 |
| **LyFormer Ours** | **0.216** | **0.465** | **0.210** | **50.0** |

Table 13: Detection Performance Comparison of AC

| Model | mAP@0.5:0.95 | AP@0.5 | $AP_S$ | FPS |
|---|---|---|---|---|
| *General baselines* | | | | |
| DETR (ResNet-50) | 0.162 | 0.365 | 0.158 | 25.5 |
| Deformable DETR (ResNet-50) | 0.171 | 0.380 | 0.164 | 24.5 |
| Swin Transformer (Swin-B) | 0.176 | 0.392 | 0.168 | 23.0 |
| DINO (ResNet-50) | 0.174 | 0.391 | 0.167 | 23.5 |
| YOLOv8s (CSPDarknet) | 0.184 | 0.398 | 0.174 | 54.5 |
| PP-YOLOE (CSPDarknet) | 0.188 | 0.405 | 0.178 | 51.5 |
| **LyFormer (Ours)** | **0.224** | **0.537** | **0.215** | **49.5** |
| *Specialized small-object backbones and detectors* | | | | |
| TinyDet | 0.194 | 0.431 | 0.174 | 52.5 |
| Bottom-heavy Tiny-Backbone | 0.205 | 0.451 | 0.184 | 58.5 |
| FocusDet | 0.235 | 0.502 | 0.215 | 47.7 |
| CRL-YOLOv5 | 0.215 | 0.461 | 0.195 | 53.5 |
| RS-TOD YOLOv8 variant | 0.246 | 0.512 | 0.225 | 50.6 |
| IYFVMNet YOLOv8-based | 0.256 | 0.532 | 0.236 | 50.0 |
| RFBNet | 0.143 | 0.341 | 0.123 | 34.7 |
| DPNet | 0.276 | 0.531 | 0.256 | 45.8 |
| *Recent SOTA detectors* | | | | |
| YOLOv10-S | 0.205 | 0.441 | 0.184 | 62.7 |
| RT-DETR R50 | 0.225 | 0.491 | 0.174 | 38.8 |
| DETA | 0.246 | 0.522 | 0.205 | 37.3 |
| SAM-DETR++ | 0.235 | 0.512 | 0.195 | 33.1 |
| *Summary total* | | | | |
| YOLOv8s (CSPDarknet) | 0.184 | 0.398 | 0.174 | 54.5 |
| **LyFormer (Ours)** | **0.224** | **0.537** | **0.215** | **49.5** |

Table 14: Detection Performance Comparison of IC

| Model | mAP@0.5:0.95 | AP@0.5 | $AP_S$ | FPS |
|---|---|---|---|---|
| *General baselines* | | | | |
| DETR (ResNet-50) | 0.180 | 0.375 | 0.172 | 26.0 |
| Deformable DETR (ResNet-50) | 0.189 | 0.395 | 0.180 | 25.0 |
| Swin Transformer (Swin-B) | 0.195 | 0.408 | 0.186 | 23.5 |
| DINO (ResNet-50) | 0.193 | 0.407 | 0.185 | 24.0 |
| YOLOv8s (CSPDarknet) | 0.197 | 0.425 | 0.189 | 53.5 |
| PP-YOLOE (CSPDarknet) | 0.203 | 0.438 | 0.193 | 50.0 |
| **LyFormer (Ours)** | **0.308** | **0.580** | **0.296** | **48.0** |
| *Specialized small-object backbones and detectors* | | | | |
| TinyDet | 0.198 | 0.416 | 0.188 | 52.5 |
| Bottom-heavy Tiny-Backbone | 0.208 | 0.437 | 0.198 | 58.5 |
| FocusDet | 0.242 | 0.508 | 0.232 | 47.7 |
| CRL-YOLOv5 | 0.218 | 0.458 | 0.208 | 43.5 |
| RS-TOD YOLOv8 variant | 0.254 | 0.533 | 0.244 | 40.6 |
| IYFVMNet YOLOv8-based | 0.264 | 0.554 | 0.254 | 47.0 |
| RFBNet | 0.150 | 0.315 | 0.140 | 34.7 |
| DPNet | 0.286 | 0.601 | 0.276 | 45.8 |
| *Recent SOTA detectors* | | | | |
| YOLOv10-S | 0.210 | 0.441 | 0.200 | 45.7 |
| RT-DETR R50 | 0.230 | 0.483 | 0.220 | 38.8 |
| DETA | 0.250 | 0.525 | 0.240 | 37.3 |
| SAM-DETR++ | 0.238 | 0.500 | 0.228 | 33.1 |
| *Summary total* | | | | |
| YOLOv8s (CSPDarknet) | 0.197 | 0.425 | 0.189 | 53.5 |
| **LyFormer (Ours)** | **0.308** | **0.580** | **0.296** | **48.0** |

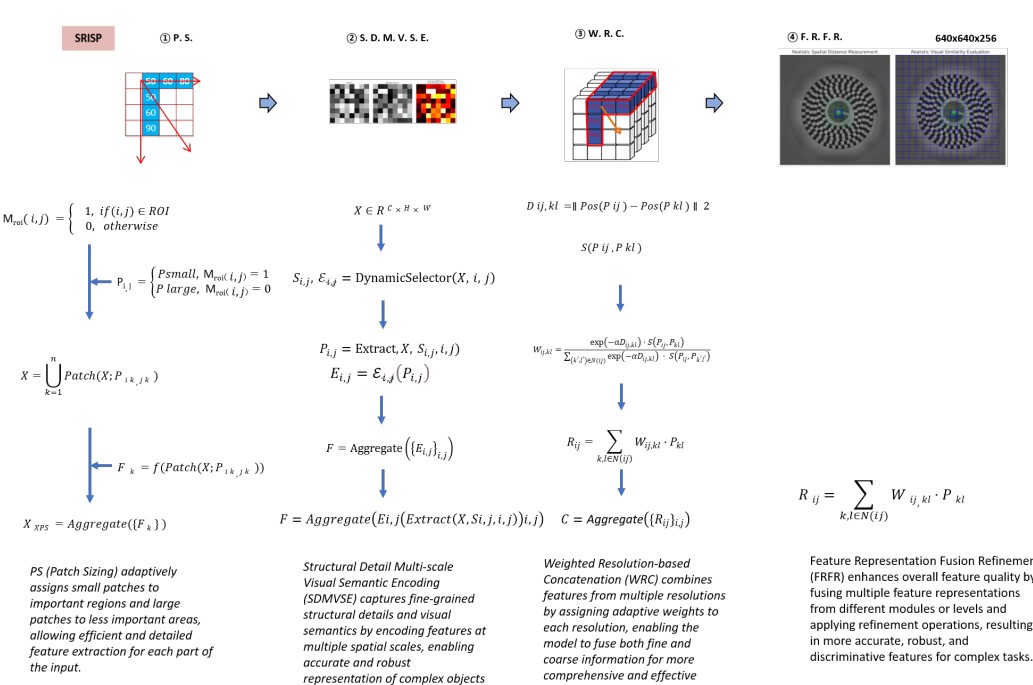

Figure 5: Overview of the SRISP module. The feature map is divided into fine-grained patches, and both spatial and visual relationships between patches are analyzed using spatial distance and visual similarity metrics. Patch-level relationships are aggregated via weighted summation, enabling precise localization and clear separation of small, densely packed objects in complex scenes.

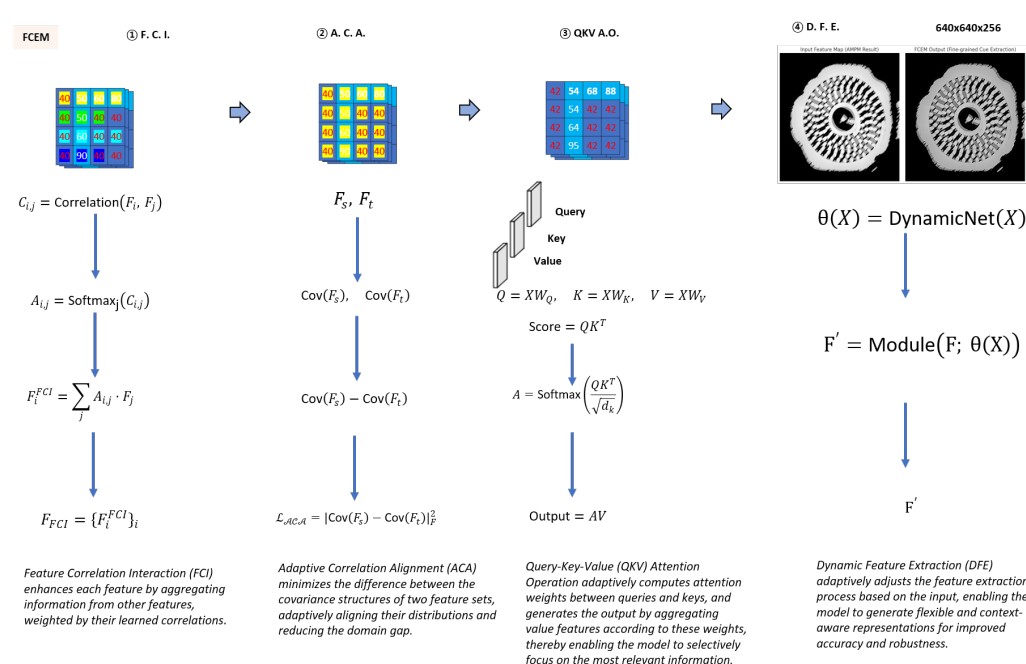

Figure 6: Overview of the FCEM module. The input feature map is processed to selectively enhance subtle and fine-grained cues using an attention-based mechanism. Query, key, and value features are generated, and attention weights are computed to highlight important visual signals. The refined output emphasizes critical features that are essential for accurate detection, especially for small or ambiguous objects in challenging visual environments.

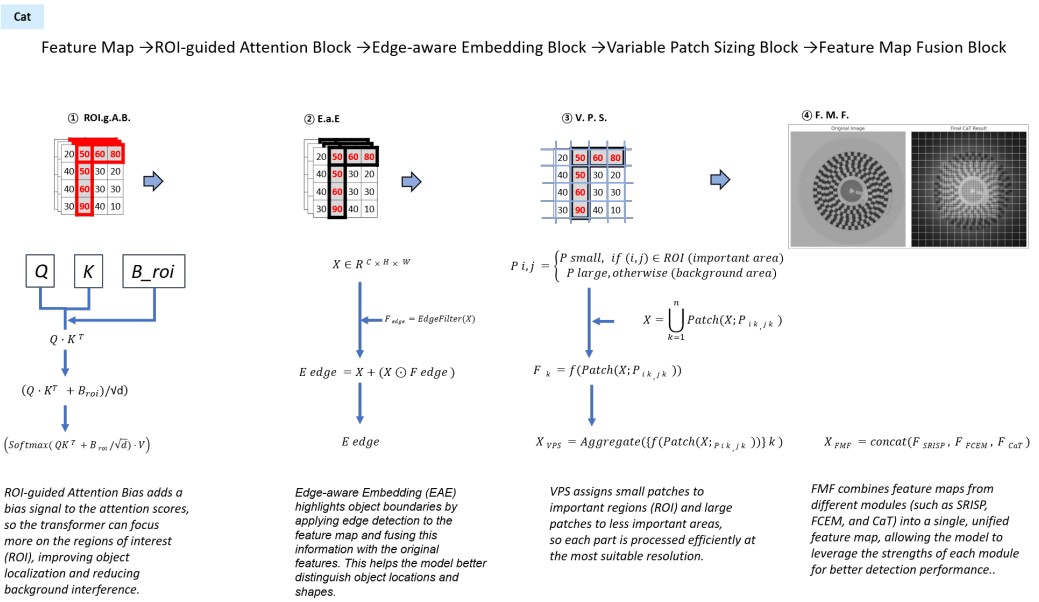

Figure 7: Overview of the CaT (Context-aware Transformer) module. The feature map undergoes ROI-guided attention, edge-aware embedding, and variable patch sizing to selectively enhance local and global contextual information. Through this multi-stage process, CaT effectively highlights object boundaries and semantic structures, enabling precise localization and robust feature representation in complex scenes.

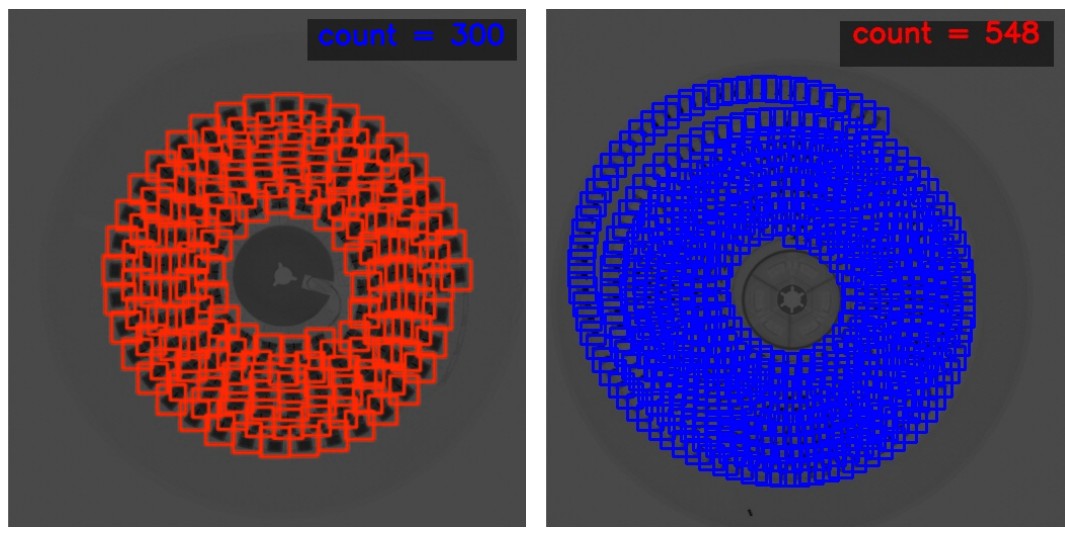

(a) Detection results LyFormer, bounding boxes, count 300     (b) Detection results LyFormer, bounding boxes, count 548

Figure 8: Chip counting results on SMT X-ray images

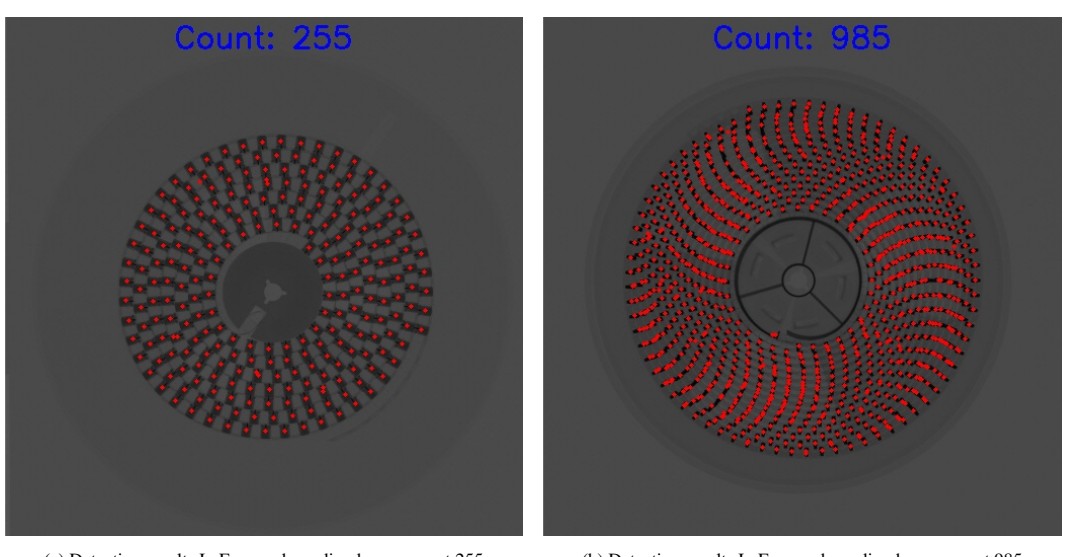

(a) Detection results LyFormer, bounding boxes, count 255     (b) Detection results LyFormer, bounding boxes, count 985

Figure 9: Chip counting results on SMT X-ray images

