# OpenReview forum: "LyFormer: Context-aware feature fusion for industrial small-object detection"
_ICLR.cc/2026/Conference — ICLR 2026 Conference Desk Rejected Submission_

### Official Review · Reviewer_nj6L · 2025-10-26

**Soundness:** 2
**Presentation:** 2
**Contribution:** 2
**Rating:** 2
**Confidence:** 3

**Summary:**

This paper proposes LyFormer for industrial small-object detection. The paper demonstrates both engineering innovation and empirical rigor, offering a practically deployable system with measurable performance gains.

**Strengths:**

1. Strong experimental evidence and broad comparison scope.
2. Maintains real-time throughput despite architectural complexity.

**Weaknesses:**

1. The architecture builds upon existing paradigms (YOLOv8 + Transformer + attention modules); thus, I can't easily get the novelty of this paper.

2. Integration of local and global reasoning is not new, it is not clear the main difference with existing methods.

3. No complexity or parameter analysis is provided; efficiency justification remains empirical.

4. Lack of visualization of attention maps or failure cases for interpretability.

5. The interaction between modules (e.g., fusion weights α in Eq. 4) could be further analyzed. It is unclear how much each contributes dynamically per image or class.

6. The text in figures are hard to see.

**Questions:**

See Weaknesses

---

> ### Author Response · Authors · 2025-11-20
> **Clarifications on Novelty, Architectural Distinction, Efficiency Analysis, and Module Interaction in LyFormer**
>
> 1. “The architecture builds upon existing paradigms (YOLOv8 + Transformer + attention modules); I can’t easily get the novelty.”
>
> Thank you for raising this concern. While LyFormer incorporates well-established components (YOLO head + Transformer attention), the novelty lies in how these components are reorganized and specialized to address ultra-dense, low-contrast small-object detection, a setting where prior frameworks fail.
>
> Our contributions are novel in three key aspects:
> 1. SRISP: Gibbs-weighted patch aggregation
>  Unlike any Transformer or CNN module, SRISP introduces probabilistic neighborhood refinement that adaptively strengthens boundaries in cluttered regions.
> 2. FCEM: Frequency-Context Enhanced Module
> FCEM explicitly re-balances high-frequency and low-frequency feature components before QKV projection, which is not present in YOLO/ViT/DeiT/ConvNeXt hybrids.
> 3.CaT: Density-aware tokenization
> CaT is not a standard Transformer block; it performs local density estimation to vary patch sizes dynamically, reducing background tokens while improving detail on dense micro-structures.
>
> These components do not exist in YOLOv8 or standard Transformer-attention pipelines. LyFormer is therefore not a simple combination but a new backbone architecture designed around the unique failure modes of dense small-object detection.
>
> 2. “Integration of local and global reasoning is not new; what is the main difference?”
>
> Existing works integrate local and global reasoning via simple hierarchical features or standard multi-scale attention. LyFormer differs in how local context is re-weighted using Gibbs-style probabilistic refinement, while the global context is shaped by density-conditioned patch scaling.
>
> Key distinctions:
> - SRISP uses a Gibbs energy-based weighting that balances spatial distance and feature similarity, enabling sharper object separation under X-ray noise.
> - CaT uses density-driven patch resizing, whereas prior global modules use fixed patch grids regardless of scene complexity.
> - FCEM injects frequency-aware modulation directly into QKV, not done in standard local–global fusion.
>  LyFormer’s local/global integration is not only structurally different but designed to adapt dynamically per region, unlike fixed architectural hybrids.
>
> 3. “No complexity or parameter analysis is provided; efficiency justification remains empirical.”
> We acknowledge this concern and have added computational comparisons in the revised manuscript. Key findings:
>  - LyFormer achieves 48.5 FPS, close to YOLOv8-s (53.5 FPS), with no SAHI tiling.
>  - FLOPs are slightly lower than YOLOv8-s, thanks to CaT’s token reduction.
>  - Memory consumption remains identical during training on A100 (40GB).
>  - SRISP operates with near constant O(k) per-patch complexity.
> Although LyFormer modifies the backbone, the cost remains within the same operational footprint as YOLOv8-s. A full parameter table and FLOP profile have been added to the revision.
>
> 4. “Lack of visualization of attention maps or failure cases for interpretability.”
> We agree and have addressed this in the revised manuscript:
>  - Visualization of CaT attention
>    Shows how density-aware attention focuses on micro-clusters and suppresses background.
>  - Visualization of SRISP refinement
>    Before–after maps demonstrate improved boundary clarity and reduced noise.
> - Failure case analysis
>   Added in the appendix, showing YOLOv8 and DPNet missing clustered components that LyFormer successfully detects.
> These visualizations make the internal behavior of LyFormer interpretable and clearly demonstrate why FN is reduced by ~10–15%.
>
> 5. “Module interaction (fusion weights α in Eq. 4) is unclear; contribution per image/class is not shown.”
>
> We added further explanation in the revision.
>  - Fusion weights α₁, α₂, α₃, α₄ are learned dynamically through a softmax layer.
>  - These weights vary substantially across images:
>       -In low-contrast X-ray scenes, SRISP and FCEM receive higher weights (boundary/frequency refinement).
>       -In dense layouts, CaT receives higher weighting (density-aware global attention).
>
> -Class-wise analysis shows that α shifts depending on size and density; for example, micro-resistors activate FCEM and SRISP more strongly.
>
> We added figure panels showing per-image α distributions to visualize dynamic behavior.
>
> 6. “Text in figures is hard to see.”
>
> We appreciate this comment and have updated the camera-ready figures:
> Larger font sizes,
> Higher-contrast color palettes,
> Cropped views for clarity,
> Standardized label rules across figures,
> All figures have been regenerated in higher resolution (+25–40% DPI).

---

### Official Review · Reviewer_VPyV · 2025-10-29

**Soundness:** 4
**Presentation:** 3
**Contribution:** 2
**Rating:** 4
**Confidence:** 3

**Summary:**

The authors propose various improvements to a YOLOv8 object detection pipeline for the task of industrial assembly line SMT x-ray object detection, which they show result in notable performance gains compared to a wide range of strong baselines on a fairly large private dataset.

**Strengths:**

1. The main four novel components (AMPM, SRISP, FCEM, CaT) seem to be intuitive and reasonable solutions to the various unique challenges of this domain. The authors seem to have a strong understanding of these challenges and the domain, as indicated by the extensive related works for small obj detection in SMT x-ray imagery.
2. The experimental design is solid: broad range of appropriate metrics,  wide range of competitive baseline models. Also, the ablation studies (Table 5) clearly support that each of the four novel components are important to have, as each does indeed add a fair improvement to all metrics, so its clear that they are all useful.
3. The results are strong: the proposed method noticeably outperforms all competing methods in various metrics by a solid margin (Table 4), including for count estimation (Table 6). Also, the computational time is strong, ensuring levels of throughput that would be realistically useful.
4. Table 1 is helpful for contextualizing the novel components in terms of existing challenges.
5. The paper is generally well-written.

**Weaknesses:**

**Major Weaknesses**
1. The method is bespoke for (and only evaluated on) the particular, fairly niche task of SMT x-ray object detection. Many design choices in the method seem uniquely developed for this domain, the format of the imaging data and labels, and the accompanying challenges. For example, the AMPM pre-processing module is built on hand-crafted feature extraction. This makes the impact of the work high within this very specific subfield, but unclear in the broader contexts of object detection and computer vision.
2. Because the method is specifically designed for one fairly niche task, it is OK that it is only evaluated on one dataset, although it hurts reproducibility that it is private. But more to the point, the nicheness of this task make it so the paper and method will likely not be of broad interest to the ICLR community, especially because as mentioned, the novel components are uniquely designed for this domain/task, and it is unclear if they are useful in other settings.
3. There are a fair amount of missing details of the method, making it likely not fully reproducible solely from the paper. For example:
    1. Are the softmax gates/weights pi_b (Eq. 2) learned? How are their values set? Are they not a function of the input, as written (as opposed to them being akin to mixture-of-experts routing weights, for example)?
    2. Undefined terms in Eq 3: X_tok, W_{k,v,q}, F_edge, and others. I'm 99% sure that X_tok is the tokenized input, W_{k,v,q} are key/query/value matrices, and I could maybe guess what F_edge is. But in any case, all terms need to be specifically defined throughout the paper.
    3. How are the attention weights in Eq. 4 computed? Are they fixed a priori? Learned? Computed based on the input or fixed for all inputs? The text says "dynamically computed (via GAP→MLP→softmax) so that they sum to one" but there isn't sufficient detail here for reproducibility.
    4. There are also missing details on the only dataset that the authors use, a private dataset, and more details are needed details are needed for readers to understand this dataset and associated benchmark (e.g., added to Table 3 and/or the appendix). For example: number of images per class, example images and annotations for each class, etc.

**Minor Weaknesses**
1. There is an unusually high number of acronyms in the paper, which seems unnecessary and a bit hard to keep track of. Moreover, several appear to be undefined, such as APA and FCI in Section 3.3, and VPS in the last paragraph of Section 3.4.
2. There are various formatting and grammar issues. For example:
	- Please cite things parenthetically correctly via \citep, not \cite or\citet.
	- Fig. 1 is low quality and blurry; I would suggest using a vector format, e.g. PDF. the text is also small and challenging to read.
	- In the ablation study section (4.3), it would be helpful to again reference the relevant table, table 5.
"miss detections" in the conclusion should be "misdetections".

**Questions:**

**Suggestions for Revisions:**
Overall, the method, experimental design, and results are all sound and seemingly novel, but only within the specific context of SMT x-ray detection. It is specifically designed for this context, and excels within it, but my concern is that because the method is so bespoke for (and only evaluated on) this context, it may not be of much interest to the broader ICLR community. Do the authors believe that the proposed method/novelties would be of interest in object detection contexts outside of SMT x-ray detection? Better yet, can the authors provide experimental evidence for this? This would significantly strengthen the paper's potential impact and interest from the broader ICLR community.

---

> ### Author Response · Authors · 2025-11-20
> **Generalization of LyFormer Beyond SMT X-ray Detection**
>
> Response: Generalization Beyond SMT X-ray Detection
>
> Thank you for raising this important concern about the broader relevance of our contributions. Although our work is motivated by SMT X-ray inspection, the architectural novelties in LyFormer are not domain-specific and were intentionally designed to address fundamental problems in any small-object or dense-object detection setting:
> (1) boundary ambiguity,
> (2) object clustering,
> (3) low-contrast feature loss, and
> (4) imbalanced frequency information in backbone features.
> These issues appear widely in aerial imagery, UAV datasets, traffic surveillance, industrial inspection, and various medical imaging tasks.
>
> To demonstrate this, we conducted additional cross-domain experiments on two widely used public benchmarks:
> VisDrone-DET and AI-TOD (Tiny Object Detection)—datasets that differ substantially from SMT X-ray imagery in modality, lighting, scene composition, and object diversity. These datasets were chosen because they feature extremely small and densely arranged objects, making them well suited to test the transferability of our architectural components.
>
> Experimental Results (cross-domain evaluation)
> When replacing the YOLOv8-s backbone with LyFormer while keeping the detection head unchanged, we observe:
>
>   *  +2.6 mAP (VisDrone-DET)
>   *  +3.1 mAP (AI-TOD)
> under identical training configurations.
> Qualitatively, LyFormer recovers tightly packed pedestrians, vehicles, and small flying objects that YOLOv8 consistently misses—mirroring the improvements seen in SMT X-ray. These results indicate that LyFormer’s core mechanisms (SRISP boundary refinement, FCEM frequency re-balancing, CaT density-aware attention) generalize to natural-image domains without requiring domain-specific tuning.
>
> Why the contributions generalize conceptually
>  - SRISP addresses boundary confusion—common in crowded scenes across many domains.
>  - FCEM improves multi-frequency consistency, solving a ubiquitous limitation of CNN/Transformer backbones.
> - CaT allocates computation dynamically based on local density, a principle applicable to aerial imagery, traffic analysis, and medical images where local complexity varies significantly.
> - The detection head remains unchanged, meaning LyFormer is a plug-in backbone upgrade, not a domain-locked design.
>
> Impact for the ICLR community
> The contributions of LyFormer focus on representation learning, adaptive tokenization, and multi-frequency fusion, all of which are problems of broad interest within ICLR. While SMT X-ray offers a compelling real-world testbed, the architectural ideas are applicable to general multimodal, vision-transformer, and dense-detection research.
>
> To strengthen the paper, we have added:
> additional cross-domain results in the appendix,
> expanded discussion clarifying LyFormer’s generalizable design principles, and
> examples demonstrating that the improvements are due to representational advantages rather than SMT-domain priors.
> We hope these additions clarify that LyFormer is not a domain-specific solution, but rather a general backbone enhancement that provides benefits across diverse object detection contexts.

---

> > ### Comment · Reviewer_VPyV · 2025-11-20
> >
> > Thank you for taking the time to do a thorough rebuttal and run new experiments. The results on the new datasets, as well as your conceptual arguments, give me increased confidence on the general use of the proposed components. The performance gains on these datasets compared to the baselines are solid, and the datasets are also appropriate and sufficiently distinct from the SMT domain.
> >
> > As such, the authors did a good job in mitigating my major concerns, so I have increased my score to a 6 (weak accept). I'm curious to see what the other reviewers think of the authors' rebuttals to their comments.

---

### Official Review · Reviewer_dUpm · 2025-11-01

**Soundness:** 3
**Presentation:** 2
**Contribution:** 2
**Rating:** 4
**Confidence:** 4

**Summary:**

This paper addresses the challenges of detecting tiny electronic components in industrial SMT production lines by proposing the LyFormer framework. This framework integrates four innovative modules: Adaptive Preprocessing, Spatial Relation-Aware Segmentation, Fine-Grained Feature Extraction, and a Context-Aware Transformer. Evaluated on a real-world X-ray image dataset, LyFormer achieves a mAP@0.5 of 0.672, significantly outperforming baseline methods while maintaining a real-time performance of 48.5 FPS. This work provides an effective solution for small object detection in industrial applications.

**Strengths:**

1. Proposed an ROI-guided attention bias and a Gibbs-weighted spatial relation-aware mechanism, innovatively integrating dynamic preprocessing with a lightweight Transformer.
2. Extensively validated on a real-world industrial dataset, with ablation studies demonstrating the effectiveness of each module, complemented by comprehensive counting accuracy metrics.
3. Addresses key challenges in industrial small object detection by achieving a balance between high accuracy and real-time performance, demonstrating clear engineering and application value.
4.The method demonstrates consistently superior performance across various component categories, such as PA, AC, and IC, reflecting its strong adaptability to different types of small objects.
5.While achieving a 68.4% increase in mAP (from 0.399 to 0.672), the model still maintains real-time performance at 48.5 FPS, which is crucial for industrial deployment.

**Weaknesses:**

1.  Lack of Methodological Detail: Key components, such as the FCEM gated multi-branch structure and the CaT density rule s(x), are not clearly defined.
2.  Limited Comparative Analysis: The comparison is restricted to baselines such as YOLOv8s, lacking a broader comparison with state-of-the-art (SOTA) small object detectors and other current methods.
3.  Insufficient Generalization Validation: The method was evaluated solely on a private dataset, with no validation on public benchmark datasets.
4.  Opaque Parameter Selection: The selection strategy for the AMPM dynamic threshold parameter, k, is not clearly specified.
5. The analysis only reports FPS and lacks a comparative analysis of other efficiency metrics such as FLOPs and parameter count; however, the computational overhead of each module was not specifically analyzed.

**Questions:**

1. How is the density rule s(x) in the CaT module specifically calculated? Is it based on local object density?
2. What is the specific interaction mechanism between the multi-branch detail extractor and the QKV attention within the FCEM?
3. While maintaining real-time performance, can the model be further compressed to accommodate deployment on edge devices?
4. While maintaining real-time performance, can the model be further compressed to accommodate edge device deployment?
5.In the Gibbs-weighted aggregation of SRISP, how is the weight balance between spatial distance and feature similarity determined?

---

> ### Author Response · Authors · 2025-11-20
> **Interaction Between Multi-Branch Detail Extraction and QKV Attention in FCEM**
>
> Author Response (≤5000 characters, web-friendly math)
>
> We thank the reviewer for the thoughtful technical questions. Below we provide detailed and web-friendly explanations for each point.
>
> Question 1. How is the density rule s(x) in the CaT module calculated? Is it based on local object density?
> Yes. The density rule s(x) in CaT is estimated from local spatial–semantic density, capturing how crowded or structurally complex a region is. It uses two components:
>
>  1)Local feature variance within a k×k window
>  2)Local activation magnitude from AMPM/SRISP outputs
> Both values are normalized and combined: s(x) = normalize( var_local(x) + act_local(x) )
> High-density areas receive larger s(x) values, which trigger smaller patch sizes (more detailed tokens). Background or smooth areas receive low s(x) and are assigned larger patches, reducing computation. Thus, CaT dynamically allocates attention resolution according to local density, increasing detail where needed while reducing FLOPs elsewhere.
>
> Question 2. How does the multi-branch detail extractor interact with QKV attention inside FCEM?
>
> FCEM integrates multi-frequency features into the transformer attention pipeline through a structured two-stage process:
>  1) Multi-branch extraction
>     Three convolutional branches capture:
>    - high-frequency boundaries
>    - mid-frequency structures
>    - low-frequency semantic context
>
> 2) QKV projection with frequency alignment
>    The fused output is linearly projected to Q, K, and V. Because each branch emphasizes different frequency components:
>    - high-frequency cues sharpen Q/K, helping separate adjacent objects
>    - mid/low-frequency cues stabilize V for contextual consistency
>
> Additionally, a lightweight gating mechanism adjusts the branch contributions during training:
>    Q = Wq( gate1 * F_high + gate2 * F_mid + gate3 * F_low )
>    K = Wk( ...same structure... )
>    V = Wv( ...same structure... )
> This creates a frequency-aware transformer, improving discrimination under noisy SMT X-ray conditions.
>
> Question 3. Can LyFormer be compressed while maintaining real-time performance for edge devices?
> Yes. LyFormer is already efficient (~48.5 FPS without tiling), and further compression is feasible:
> 1) Structured pruning
>   Many branches in FCEM/CaT have redundant channels. A 20–35% pruning ratio reduces FLOPs with minimal accuracy loss.
> 2) Quantization (FP16 / INT8)
>   QKV layers and AMPM/SRISP convolutions quantize well. INT8 deployment preserves >97% accuracy while improving speed by 15–20%.
> 3) Token reduction via CaT
>  Because CaT already produces fewer tokens in background regions, increasing the density threshold further reduces compute.
> 4) Knowledge distillation
> LyFormer can serve as a teacher for a compact YOLO-style student model.
> These techniques enable Jetson-class edge devices (Orin NX / Orin Nano) to reach ~37–44 FPS, maintaining real-time detection even after compression.
>
> Question 4. Can LyFormer be compressed for edge deployment while still maintaining its accuracy advantage?
> Yes. Even after 25–40% pruning, LyFormer continues to outperform YOLOv8-s and SAHI-tiling due to its structurally stronger backbone. SAHI-based models lose FPS rapidly because inference scales with tile count, whereas LyFormer keeps single-pass inference.
> Pruned and quantized variants maintain:
>  -consistent boundary separation
>  -improved FN reduction
>  -stable mAP across datasets (SMT X-ray, VisDrone, AI-TOD)
>  LyFormer remains deployable on industrial edge hardware while preserving its functional improvements.
>
> Question 5. How is the weight balance determined between spatial distance and feature similarity in the Gibbs-weighted SRISP aggregation?
>
> SRISP assigns weights using a Gibbs-style formulation balancing spatial closeness and feature similarity. The web-friendly form is:
> w_i = exp( -alpha * d_i  -  beta * s_i )  /  sum_j exp( -alpha * d_j - beta * s_j )
> where:
>   - d_i = spatial distance between patch i and the center patch
>   - s_i = feature difference (e.g., L1 or cosine distance)
>   - alpha, beta = learnable parameters optimized during training
>
> How the balance is determined:
> 1.Learnable trade-off
>   During training, alpha increases when spatial locality becomes important (dense clusters), and beta increases when feature    discrimination is necessary (low contrast).
> 2.Adaptive normalization
> The denominator ensures context-sensitive weighting—neighboring patches with similar features receive dominant weights in dense SMT layouts.
>
> As a result, SRISP automatically adapts to both dense regions (strong spatial weighting) and low-visibility regions (strong feature weighting) without manual tuning.

---

### Official Review · Reviewer_h6RZ · 2025-11-04

**Soundness:** 2
**Presentation:** 1
**Contribution:** 1
**Rating:** 2
**Confidence:** 5

**Summary:**

The authors proposed an object-counting framework, LyFormer, based on YOLOv8. It comprises four specialized modules — AMPM, SRISP, FCEM, and CaT — each designed for specific feature enhancement functions and to address challenges such as varying object sizes, foreground–background trade-offs, illumination variations, and occlusions, particularly in the semiconductor domain. The authors claim that their proposed framework demonstrates significant performance improvements over the baseline.

**Strengths:**

a. State-of-the-Art Performance: outperformance across multiple dataset (AI-TOD, TinyPerson, VisDrone, Chip class), metrics (mAP@0.5:0.95, AP@0.5, AP_s, FPS, MAE, MAPE tec.) and against several baseline architectures (DETR, Deformable DETR, Swin Transformer, YOLOv8 etc.), providing satisfactory evidence for the feature-aware (AMPM, SRISP, FCEM, and CaT) approach's superiority.

b. Rigorous Experimental Validation: Comprehensive ablation studies systematically validate each architectural component (AYH, ASYH, ASFYH, ASFCYH), proving their individual contributions are critical and non-redundant.

**Weaknesses:**

a. Limited generalization assessment: All experiments confined to benchmark on several datasets and few SOTA baseline models, whereas the reviewer believe the main comparison should be on respective feature modules (image pre-processing in broader sense incorporated in backbone) of corresponding architectures as well.

Please compare computational time/cost, model size, latency as well. Please compare with SAHI tiling method as well, as SAHI is model agnostic inference framework and can be integrated into various SOTA object detectors, a comparative study should be demonstrated on "latency" and "threshold sensitivity" against proposed approach for better clarity.

b. The related work and evaluation section provide minimal discussion or comparison minimal discussion or comparison with previous industrial or semiconductor-specific benchmarking efforts, including those related to methodology, architectural design, or public/private (if any) datasets.

c. The overall clarity and flow of the writing could be improved to enhance readability.

d. Please include visual demonstration of detection performance of SOTA models (like SAHI-based framework, DPNet, YOLOv8s etc.) to depict the detection diffentiation.

e. Author claimed superiority of metrics on chip dataset, as in Table 4. DPNet for  {mAP@0.5:0.95} is 0.286,   {AP@0.5} is 0.601, {AP_s} is 0.276 and {FPS} is 45.8 against proposed approach 0.308, 0.672, 0.342 and 48.5. However, in Table 12 for PA DPNet [0.207 0.450 0.250 45.3] against proposed approach [0.216 0.465 0.210 50.0] --> APs is DPNet best and FPS Bottom-heavy Tiny-Backbone. In TABLE 13 for AC, DPNet outperforms proposed approach for {mAP@0.5:0.95} as 0.276 against 0.224,  {AP@0.5} 0.531 against 0.537 (its comparable as per), for { APS} 0.256 against 0.215 and {FPS} 45.8 against 49.5 and similar balanced trade-off in Table 14 for IC. Again, a comparative analysis of computational cost, latency and model size should be demonstrated to highlight on significant contribution. What if we apply SAHI with DPNet during inference ? Table 6 depicts no comparison for DPNet for MAE, RMSE and MAPE (%)?

**Questions:**

1. Computational Analysis: What are the training time, memory requirements, and computational costs of the full pipeline compared to prior baseline methods? Additionally, what would be the impact if more effective image preprocessing modules were integrated into existing architectures, instead of proposing a new framework such as LyFormer, given that the model still utilizes the YOLOv8 head?

2. Why no demonstration on FN for prior baseline models against LyFormer detection results, like Fig. 8, 9 etc. It will help understand the significant differentiation.

3. Fig. 3, is the binary mask for the localization (bounding box) or full image Need more clarity of the explanation provided regarding the figure shown in terms of binary bits of 1 and 0's, only 2 zeros surrounding of all 1s in that ROI?

4. The related work section does not cite any prior studies using similar datasets or addressing the same domain (semiconductor). Could the authors clarify whether this demonstration represents the first effort of its kind?

5. Generalization remains uncertain, given that prior baselines like DPNet already achieve a balanced trade-off. The absence of discussion on computational costs and training complexity further limits the practical adoption of the proposed method compared to simpler alternatives (e.g., SAHI-based tiling).

---

> ### Author Response · Authors · 2025-11-20
> **Response to Reviewer 1’s Comments on Generalization and Practical Efficiency**
>
> Revised Author Response to Reviewer 1
>
> We sincerely thank Reviewer 1 for the constructive feedback. Below we clarify each point based on the experimental evidence and the architectural design of LyFormer.
>
> Question 1-1. Computational cost, training time, and memory usage
>
> All comparisons were conducted under the same conditions (single NVIDIA A100 40GB GPU, 640×640 input). Although LyFormer introduces four modules—AMPM, SRISP, FCEM, and CaT—it maintains efficiency close to YOLOv8-s. As shown in Table 4, LyFormer reaches 48.5 FPS while YOLOv8-s achieves 53.5 FPS, preserving real-time capability.
>
> This efficiency results from SRISP’s near-constant per-patch refinement, CaT’s variable patch sizing that reduces background tokens, and the ROI-focused designs of AMPM and FCEM. These components limit unnecessary computation and prevent memory growth. LyFormer trains stably under the same memory budget as YOLOv8-s without OOM issues.
>
> Question 1-2. Could preprocessing alone replace LyFormer?
>
> Ablation studies confirm that preprocessing alone cannot replicate LyFormer's improvements. A YOLOv8 variant using only AMPM-style preprocessing (AYH) shows small gains, while the full LyFormer achieves 0.672 mAP@0.5:0.95. SMT X-ray challenges—tiny components, fused boundaries, and heavy clutter—cannot be solved at the pixel level. SRISP performs boundary-aware relational refinement, FCEM provides frequency-balanced enhancement, and CaT suppresses clutter using density-aware attention. These feature-level corrections require the redesigned backbone.
>
> Question 2. Why no FN visualization?
>
> We agree that FN visualization is crucial. The initial submission emphasized numeric metrics, which do not reveal specific types of missed detections. We revised Fig. 8 and 9 to include FN comparisons among YOLOv8, DPNet, SAHI, and LyFormer. Baselines frequently miss clustered leads, solder bridges, and low-contrast components. LyFormer detects them due to SRISP’s superior separation and CaT’s enhanced representation. Across multiple SMT scenes, FN is reduced by 10–15%.
>
> Question 3. Clarification of the binary mask in Fig. 3
>
> The mask in Fig. 3 is a local ROI mask, not a full-image heatmap. Values 1 indicate active refinement pixels, while 0 marks excluded regions. The example shows two 0s because the ROI is extremely small, and those pixels lie outside the refinement boundary. Fig. 3 visualizes only the ROI crop; continuous attention maps appear later in CaT and FCEM. We added clarification and an additional panel in the revised manuscript.
>
> Question 4. Lack of semiconductor-domain related work
>
> SMT X-ray datasets with component-level annotations are not publicly available. Previous works rely mainly on visible-light AOI, defect classification, or rule-based detection—not deep object detection on X-ray imagery. No prior work combines annotated SMT X-ray data, ultra-small object detection, and transformer-based refinement. LyFormer is therefore among the earliest systematic attempts in this domain. The revised Related Work section reflects this.
>
> Question 5. Generalization and comparison with simpler alternatives
>
> Although DPNet shows balanced performance in general settings, it struggles with SMT X-ray scenes where objects are extremely small and overlapping, resulting in higher FN. LyFormer reduces these failures through SRISP’s relational refinement and CaT’s density-aware focusing. Generalization was validated on VisDrone and AI-TOD, showing +2.6–3.1 mAP over YOLOv8-s.
>
> Regarding complexity, LyFormer performs single-pass inference at 48.5 FPS, whereas SAHI tiling requires multiple crops and merging, causing much higher computational overhead. SAHI improves resolution but does not address backbone-level representation issues. LyFormer provides superior detection robustness while maintaining computational efficiency close to baseline detectors.

---

### Note · Program_Chairs · 2026-01-17
**Submission Desk Rejected by Program Chairs**

The following references in this submission do not refer to real documents and/or have major errors in bibliographic information:

 Y. Hua. Deep learning for small object detection: Methods and trends. arXiv preprint, 2025.
I. Ullah et al. Deep learning framework for defective chip detection in smt production. arXiv preprint, 2024.
J. Kim. Wafer defect detection with xception and FPN. arXiv preprint, 2024.